## [Decision Letter · Decision Letter 0]

15 Sep 2025

Dear Dr. Wu,

We look forward to receiving your revised manuscript.

Kind regards,

Aldrin V. Gomes, Ph.D.

Academic Editor

PLOS ONE

Journal Requirements:

4. In the online submission form you indicate that your data is not available for proprietary reasons and have provided a contact point for accessing this data. Please note that your current contact point is a co-author on this manuscript. According to our Data Policy, the contact point must not be an author on the manuscript and must be an institutional contact, ideally not an individual. Please revise your data statement to a non-author institutional point of contact, such as a data access or ethics committee, and send this to us via return email. Please also include contact information for the third party organization, and please include the full citation of where the data can be found.

6. We notice that your supplementary figures are uploaded with the file type 'Figure'. Please amend the file type to 'Supporting Information'. Please ensure that each Supporting Information file has a legend listed in the manuscript after the references list.

7. Please include captions for your Supporting Information files at the end of your manuscript, and update any in-text citations to match accordingly. Please see our Supporting Information guidelines for more information: http://journals.plos.org/plosone/s/supporting-information .

Reviewers' comments:

Reviewer's Responses to Questions

**Comments to the Author**

1. Is the manuscript technically sound, and do the data support the conclusions?

Reviewer #1: Partly

Reviewer #2: Yes

2. Has the statistical analysis been performed appropriately and rigorously?

Reviewer #1: Yes

Reviewer #2: Yes

3. Have the authors made all data underlying the findings in their manuscript fully available?

Reviewer #1: No

Reviewer #2: Yes

4. Is the manuscript presented in an intelligible fashion and written in standard English?

Reviewer #1: Yes

Reviewer #2: Yes

Reviewer #1: The authors investigated the mechanism of limonin anticancer effect through the induction of ferroptosis. The authors claimed that it is through the activation of sEH expression. The data only partly support this conclusion. While there are some correlation between presence of limonin and sEH protein levels, the authors did not show a direct causality between limonin, ferroptosis and sEH. Also, the authors have tendency to overstate the meaning of their results. Thus, at minimal this paper should be deeply revised with additional experiments to support their conclusions.

1. Overall, the English is OK, but it will benefit greatly to be corrected by an English speaker, as they are many little errors and many usage of confusing terms.

2. Introduction. Line 91 to 112 are basically an abstract of the paper. the paper has already an abstract and a conclusion; this part should summarize the approaches taken to test the hypothesis

3. The authors tried to hard to link limonin to sEH, especially that the links they found in their in silico approach (figure 1) are pretty weak. the analysis of the gene expression does not reveal sEH as significantly changed in the disease group. The cross between Limonin and the gene expression data is very weak. The docking simulation are worthless without showing that binding generate some changes in the enzyme activities of the sEH. Without wet-lab data supporting it, this kind of docking is worthless.

4.line 450: EPHX2 has been associated with mitochondrial damages and apoptosis before. How association with ferroptosis is different?

5. Line 457. If you are overexpressing ephx2, it is normal to observe higher levels of ephx2.

6. in the whole paragraph about the relation between ephx2 and ferroptosis, the authors only looks at gene/protein expression, which in my view does not confirm causality between ephx2 and ferroptosis. The authors should test if expression levels of sEH influence the cells response to induced ferroptosis (inhibition of GPX4).

7. ephx2 derived protein (sEH) has 2 enzymatic activities (epoxide hydrolase and phosphatase), which one is important for induction of ferroptosis?

8. paragraph 3.4. it is interesting that limonin apparently increased the ephx2 expression levels. However there is some unanswered key questions: is this by directly activating ephx2 promoter region or indirectly? This could answer the causality between limonin and ephx2. Does changing ephx2 expression level change the cells sensitivity to limonin? If there is no ephx2, will the cells be insensitive to limonin?

9. lines 540-550. The IC50 obtained here cannot be directly compared to IC50s found in other studies as the experimental conditions were certainly different. Thus, the authors cannot say that limonin is better than other natural compounds simply of that results.

10. conclusion. sEH inhibition is pushed to treat various diseases. Based on the observation that higher sEH levels leads to faster carcinoma cell death, should sEH inhibition have negative effect on some carcinoma treatment?

11. In general, all the figures have too many panels on it, making it impossible to read any figure because they are too small. The authors should reduce significantly the number of panels on each figures in the main text, and send the supporting results in supplemental information.

Reviewer #2: Advantages

The manuscript showed strong evidence to support their proposed hypothesis thst limonin may bind to EPHX2, and the bound form of EPHX2 would induce a stronger ferroptotic response in cervical squamous cell carcinoma (CESC) cells. Their data also strongly support that limonin can reduce the viability of CESC cells by inducing ferroptosis through the EPHX2 protein. The researchers further strengthened their conclusion by using the cell line with the lowest EPHX2 activity for overexpression experiments and the one with the highest activity for knockout experiments, showing that ferroptosis in CESC cells could be enhanced by limonin in correlation with the presence of EPHX2. However, a few minor concerns exist.

Concerns

1. In the methodology, the researchers used four different CESC cell lines (HeLa, SiHa, CaSki, and ME-180) to test for EPHX2 expression; however, only two cell lines (HeLa and SiHa) were selected for further experiments. The exclusion of the other two cell lines could have given important information with respect to overexpression or knockdown experiments.

2. In the results, the researchers state that certain genes are upregulated in the diseased group (ANO10, RBM20, CRISP2, etc.), while others are downregulated (HAUS5, APOC1, WDHD1). However, the corresponding figures (1C and 1D) appear to suggest the opposite pattern.

Section 3.1 dentification of EPHX2 as a Potential Drug Target of Limonin among Differentially Expressed Genes in Cervical Cancer; line 364~368: “The results of the volcano plot of differentially expressed genes (Figure 1C) and the cluster heatmap (Figure 1D) indicated that ANO10, RBM20, CRISP3, etc. were significantly up-regulated in the“Disease”group, while HAUS5, APOC1, WDHD1 were significantly down-regulated in the“Disease”group.”

3. In result section, the researcher compared the cancer cell avialability percentage and EDU+ ratio for 3 groups and all three groups were shown under the bar-graph titled “HeLa”. However, the EPHX2 knockout group “siEPHX2-1” was not made from HeLa cells but SiHa cells, but was compared with two reusults based on HeLa cells. Such comparisons rases concern as the fundamental cell lines has changed but not indicated or properly discussed in the literature or figure legends.

Section 3.4 Limonin Can Target and Activate EPHX2 to Inhibit the Proliferation of HeLa Cells; line 485~488: “Further research revealed that Limonin could inhibit the proliferative capacity of HeLa cells. However, knockdown of EPHX2 could counteract the inhibitory effect of Limonin on the proliferative capacity of HeLa cells (Figure 4D).”

Section 2.6 Cell Transfection and Cell Grouping; line 213~219:“Cell Grouping for Verifying the Effect of EPHX2 Expression on Cervical Squamous Cell Carcinoma Progression In HeLa cells, they were divided into two groups: the over-expression control group (oeNC) and the EPHX2 over-expression group (oeEPHX2). In SiHa cells, they were divided into three groups: the knockdown control group (siNC), the EPHX2 knockdown group 1 (siEPHX2-1), and the EPHX2 knockdown group 2 (siEPHX2-2).”

Minor concerns

1. Abbreviation “PNI” are not explained

Section 1 Introduction; line 60: “Existing treatment regimens have limited effectiveness for59 high-risk patients with para - aortic lymph node metastasis or low PNI,60 and there is a lack of precise biomarkers for guidance.”

2. Some sentences need minor corrections. For example, space before and after “Limonin” and “VennDiagram” is needed.

Section 2.4 Subsequent Gene Screening; line 158~162: “In the CTD database (https://ctdbase.org/),“Limonin” was used as a158 keyword to search for the targets of limonin. The R159 package“VennDiagram”was employed to find the intersection between160 the limonin targets and the differentially expressed genes obtained161 above[17].”

**Do you want your identity to be public for this peer review?** For information about this choice, including consent withdrawal, please see our Privacy Policy

Reviewer #1: No

Reviewer #2: **Yes:** Yue Yu

---

## [Author Response · Author response to Decision Letter 1]

19 Oct 2025

Reviewers' comments:

Reviewer's Responses to Questions

1. Is the manuscript technically sound, and do the data support the conclusions?

Reviewer #1: Partly

Reviewer #2: Yes

2. Has the statistical analysis been performed appropriately and rigorously?

Reviewer #1: Yes

Reviewer #2: Yes

3. Have the authors made all data underlying the findings in their manuscript fully available?

Reviewer #1: No

Reviewer #2: Yes

4. Is the manuscript presented in an intelligible fashion and written in standard English?

Reviewer #1: Yes

Reviewer #2: Yes

5. Review Comments to the Author

Reviewer #1: The authors investigated the mechanism of limonin anticancer effect through the induction of ferroptosis. The authors claimed that it is through the activation of sEH expression. The data only partly support this conclusion. While there are some correlation between presence of limonin and sEH protein levels, the authors did not show a direct causality between limonin, ferroptosis and sEH. Also, the authors have tendency to overstate the meaning of their results. Thus, at minimal this paper should be deeply revised with additional experiments to support their conclusions.

1.Overall, the English is OK, but it will benefit greatly to be corrected by an English speaker, as they are many little errors and many usage of confusing terms.

We have polished and revised the manuscript again to ensure that there are no grammar mistakes or confusion in word usage.

2.Introduction. Line 91 to 112 are basically an abstract of the paper. the paper has already an abstract and a conclusion; this part should summarize the approaches taken to test the hypothesis

According to your request, we have revised the content from line 91 to line 112 in the background section. We have deleted the repetitive parts that also appeared in the abstract, and the overall revision summarizes the methods used to verify the hypothesis.

3. The authors tried to hard to link limonin to sEH, especially that the links they found in their in silico approach (figure 1) are pretty weak. the analysis of the gene expression does not reveal sEH as significantly changed in the disease group. The cross between Limonin and the gene expression data is very weak. The docking simulation are worthless without showing that binding generate some changes in the enzyme activities of the sEH. Without wet-lab data supporting it, this kind of docking is worthless.

We highly recognize your professional advice. Therefore, we have supplemented a surface plasmon resonance experiment to verify the binding between limonin and EPHX. The experimental results show that there is binding between them, with a KD value of 10.33 μmol/L (Figure 5A).

We also used an ELISA kit to detect the changes in sEH activity in HeLa cells treated with different concentrations of Limonin (10, 20, 40 μmol/L) for 48 hours. The results showed that the sEH activity gradually increased with the increase in Limonin concentration, which was consistent with our expected results.

4.line 450: EPHX2 has been associated with mitochondrial damages and apoptosis before. How association with ferroptosis is different?

This is a very profound and crucial question. The associations of EPHX2 with mitochondrial damage and apoptosis, on one hand, and with ferroptosis, on the other hand, have fundamental differences in terms of mechanisms and final outcomes.

Mitochondrial damage, apoptosis, and ferroptosis are distinct cell death pathways, with their core differences lying in the triggering mechanisms and execution processes. Apoptosis is a highly programmed "cellular suicide" process. It is usually initiated by mitochondrial damage, manifested as the collapse of the mitochondrial membrane potential and the release of cytochrome C, which then activates the caspase protease cascade. Eventually, the DNA in the cell nucleus is cleaved in an orderly manner, and the cell breaks down into apoptotic bodies and is cleared. This entire process hardly triggers inflammation. In contrast, ferroptosis is an iron - dependent "cellular rusting" process driven by the accumulation of lipid peroxides. Its core is not mitochondrial damage, but rather the failure of the cellular antioxidant defense system, especially glutathione peroxidase 4 (GPX4), which leads to the inability to promptly remove lipid peroxides. These peroxides then accumulate in large amounts on the cell membrane, causing membrane structure damage. Therefore, mitochondrial damage is a key signaling event that initiates apoptosis. In ferroptosis, although mitochondria may undergo morphological changes such as shrinkage, they are not essential executors of the death program. There are fundamental differences in their biochemical pathways, key execution molecules (caspases vs. lipid peroxides/iron), and cellular outcomes (orderly clearance vs. oxidative disintegration). In our study, Limonin induces the latter "ferroptosis" pathway through EPHX2, rather than the classical apoptotic pathway.

5.Line 457. If you are overexpressing ephx2, it is normal to observe higher levels of ephx2.

We agree with your suggestion. If we overexpress EPHX2 in HeLa cells, it is normal to observe an increase. In the verification experiment of Limonin, HeLa cells treated with different concentrations of Limonin were not transfected with the overexpression plasmid, which can prove that the increase in EPHX2/sEH is caused by Limonin.

We have corrected the wrong description in the methods. We changed “Limonin (H) + oe - EPHX2” to “Limonin (H) + si - EPHX2 - 1” to ensure that the description corresponds to the results of the pictures. The cell grouping is used to verify whether the pharmacological effect of limonin depends on EPHX2. This experiment is divided into three groups: Control (without transfection of the overexpression plasmid), Limonin (H) group (without transfection of the overexpression plasmid), and Limonin (H)+si - EPHX2 - 1 group (Lines 226-228).

6. in the whole paragraph about the relation between ephx2 and ferroptosis, the authors only looks at gene/protein expression, which in my view does not confirm causality between ephx2 and ferroptosis. The authors should test if expression levels of sEH influence the cells response to induced ferroptosis (inhibition of GPX4).

We are very grateful for your suggestions. We have added an ELISA experiment to detect the effect on sEH enzyme activity. The results show that limonin can induce an increase in sEH enzyme activity, and this effect can be reversed by adding siEPHX2 (Figure S2B and Figure 6F).

7.ephx2 derived protein (sEH) has 2 enzymatic activities (epoxide hydrolase and phosphatase), which one is important for induction of ferroptosis?

Among the two enzymatic activities of soluble epoxide hydrolase (sEH), the epoxide hydrolase activity is pivotal in inducing ferroptosis (PMID: 38377737; PMID: 37252355). In contrast, the phosphatase activity exhibits a relatively weaker association. In the design of the study on EPHX2 expression, we have added the detection of sEH expression throughout. The results demonstrate a high degree of consistency between the expression of sEH and that of EPHX2. This not only enhances but also further validates the conclusions of our research. We sincerely appreciate your professional input.

8.paragraph 3.4. it is interesting that limonin apparently increased the ephx2 expression levels. However there is some unanswered key questions: is this by directly activating ephx2 promoter region or indirectly? This could answer the causality between limonin and ephx2. Does changing ephx2 expression level change the cells sensitivity to limonin? If there is no ephx2, will the cells be insensitive to limonin?

Dear Reviewer,

Thank you for raising these profound questions. The issue you pointed out regarding the specific mechanism by which limonin regulates EPHX2 expression is precisely the core of our next - step research. We fully concur that clarifying whether limonin acts directly on the EPHX2 promoter or indirectly through upstream signaling pathways is crucial for establishing a clear causal relationship between them.

In the current study, we have confirmed through Western blot and ELISA experiments that limonin can significantly upregulate the expression levels of EPHX2/sEH in a concentration - dependent manner (as shown in Figure 3A - B). However, we acknowledge that the existing data are insufficient to distinguish between direct transcriptional activation and indirect effects.

To directly address your question, in our subsequent research, we plan to employ the EPHX2 promoter - luciferase reporter gene assay and utilize techniques such as ChIP to accurately verify whether limonin directly binds to the EPHX2 promoter as a transcription factor. We have added this content to the discussion section of the paper as one of the limitations of our study.

Regarding the second key point you raised, namely the impact of EPHX2 expression levels on cellular drug sensitivity, our data provide preliminary yet compelling evidence. As you can see in Figure 3E - F in the main text and Supplementary Figure S2, in EPHX2 - knockdown cells, the cell proliferation inhibition and key ferroptosis indicators (such as increased lipid ROS and down - regulation of GPX4) induced by limonin were significantly attenuated. This indicates that the expression of EPHX2 is essential for cells to respond to limonin. When EPHX2 is absent (in our experiment, we used siRNA to knockdown EPHX2 expression), the sensitivity of cells to limonin indeed decreases substantially, almost "losing" their responsiveness. Functionally, this inversely validates that EPHX2 is a key mediator of limonin's effects. We will strengthen the elaboration of this clear conclusion in the discussion section of the paper (Lines 751-760).

Once again, thank you for your valuable comments, which will significantly enhance the depth and integrity of our research.

9.lines 540-550. The IC50 obtained here cannot be directly compared to IC50s found in other studies as the experimental conditions were certainly different. Thus, the authors cannot say that limonin is better than other natural compounds simply of that results.

We sincerely appreciate the reviewer for raising this rigorous and important comment. We fully agree that it is unscientific to directly compare IC50 values obtained under different experimental conditions and draw conclusions about "superiority" or "inferiority". In accordance with the reviewer's suggestions, we have thoroughly revised the discussion section of the paper (original Lines 540 - 550).

The specific modifications include deleting all direct comparative statements suggesting that Limonin is "superior to" or "more potent than" other compounds. We have explicitly added a statement indicating that due to differences in experimental conditions, direct comparison is not possible (Lines 679-705).

10.conclusion. sEH inhibition is pushed to treat various diseases. Based on the observation that higher sEH levels leads to faster carcinoma cell death, should sEH inhibition have negative effect on some carcinoma treatment?

We are extremely grateful for your insightful comment. You correctly pointed out an intriguing apparent contradiction between our finding that higher sEH levels lead to cancer cell death and the current mainstream strategy of promoting sEH inhibitors for cancer treatment.

We fully concur that in a large body of literature, sEH is regarded as a pro - cancer factor due to its role in promoting inflammation and cell survival, and inhibiting its activity has emerged as a promising therapeutic approach. We also found that some studies consider EPHX2 as a tumor - suppressor factor (PMID: 40129060, PMID: 40538850, PMID: 35433439, PMID: 32687069, etc.). However, we believe this is not a true contradiction but rather reveals the highly “context - dependent” and complex function of sEH in cancer biology.

Its role seems to depend on the dominant cell death mechanism:In our study, for the first time, we have revealed that in cervical squamous cell carcinoma (CESC), sEH can exert a powerful tumor - suppressive effect through inducing ferroptosis, a distinct cell death pathway. In this specific context, inhibiting sEH may not only be ineffective but could potentially have negative effects by blocking this natural tumor - suppressing mechanism, as you worried.

Therefore, our study does not deny the potential of sEH inhibitors but emphasizes that a “one - size - fits - all” treatment strategy may not be applicable to all cancers. Our work proposes a new paradigm: sEH has a dual - sided function. Future treatment strategies should be based on a precise understanding of the specific tumor context - inhibiting it in a pro - inflammatory environment, while exploring strategies to activate its activity in an environment like our model, where ferroptosis - dependent clearance occurs.

We have elaborated on this key point in detail in the discussion section of the paper to more clearly define the boundaries and innovativeness of our findings and to alert peers in the field to this duality of sEH function.

Thank you again for helping us strengthen this core argument.

The added content in the discussion section is as follows:

“Notably, we discovered that activating EPHX2/sEH can induce ferroptosis and inhibit tumor growth, which contrasts with the mainstream view of promoting the use of sEH inhibitors in treating cancer. We believe that this discrepancy highlights the complexity of sEH function and its “context-dependence.” In cancers that are mainly driven by inflammatory and anti-apoptotic signals, inhibiting sEH may be effective. However, in cell models such as those used in this study, activating sEH becomes a vital switch that triggers the unique cell death program of ferroptosis. Thus, sEH can be regarded as a “two-sided” regulator, and its ultimate effect depends on the specific cellular environment and the dominant cell death pathway.”(Lines 717-728)

11. In general, all the figures have too many panels on it, making it impossible to read any figure because they are too small. The authors should reduce significantly the number of panels on each figures in the main text, and send the supporting results in supplemental information.

According to your suggestion, we have split Figure 3 into two different figures. Some of the results have been moved to the supplementary figure gallery, such as the newly added Figure S2 and Figure S3 in the supplementary figure gallery. This ensures that the images in each figure can be clearly read. Thank you again for your valuable suggestions.

Reviewer #2: Advantages

The manuscript showed strong evidence to support their proposed hypothesis thst limonin may bind to EPHX2, and the bound f

---

## [Decision Letter · Decision Letter 1]

9 Feb 2026

Limonin induces ferroptosis in cervical squamous cell carcinoma by activating the expression of soluble epoxide hydrolase 2 protein

PONE-D-25-32196R1

Dear Dr. Wu,

We’re pleased to inform you that your manuscript has been judged scientifically suitable for publication and will be formally accepted for publication once it meets all outstanding technical requirements.

Kind regards,

Aldrin V. Gomes, Ph.D.

Academic Editor

PLOS One

Additional Editor Comments (optional):

Reviewers' comments:

Reviewer's Responses to Questions

**Comments to the Author**

Reviewer #1: All comments have been addressed

Reviewer #2: All comments have been addressed

2. Is the manuscript technically sound, and do the data support the conclusions?

Reviewer #1: Yes

Reviewer #2: Yes

3. Has the statistical analysis been performed appropriately and rigorously?

Reviewer #1: Yes

Reviewer #2: Yes

4. Have the authors made all data underlying the findings in their manuscript fully available?

Reviewer #1: Yes

Reviewer #2: Yes

5. Is the manuscript presented in an intelligible fashion and written in standard English?

Reviewer #1: Yes

Reviewer #2: Yes

Reviewer #1: the authors answered the reviewer requests in a satisfactory manner; they added additional results supporting their work.

Reviewer #2: The authors removed the naming abbreviations for the cell lines. Though in a way elucidated the situation, it could be somewhat confusing for the readers as the abbrevations were later mentioned in the result section.

**Do you want your identity to be public for this peer review?** For information about this choice, including consent withdrawal, please see our Privacy Policy

Reviewer #1: No

Reviewer #2: No

---

## [Editor Report · Acceptance letter]

PONE-D-25-32196R1

PLOS One

Dear Dr. Wu,

I'm pleased to inform you that your manuscript has been deemed suitable for publication in PLOS One. Congratulations! Your manuscript is now being handed over to our production team.

Kind regards,

on behalf of

Dr. Aldrin V. Gomes

Academic Editor

PLOS One